# The Effects of Insulin on Immortalized Rat Schwann Cells, IFRS1

**DOI:** 10.3390/ijms22115505

**Published:** 2021-05-23

**Authors:** Tomokazu Saiki, Nobuhisa Nakamura, Megumi Miyabe, Mizuho Ito, Tomomi Minato, Kazunori Sango, Tatsuaki Matsubara, Keiko Naruse

**Affiliations:** 1Department of Pharmacy, Aichi Gakuin University Dental Hospital, Nagoya 464-8651, Japan; saiki@dpc.agu.ac.jp; 2Department of Internal Medicine, School of Dentistry, Aichi Gakuin University, Nagoya 464-8651, Japan; mmiyabe@dpc.agu.ac.jp (M.M.); i-mizuho@dpc.agu.ac.jp (M.I.); matt@dpc.aichi-gakuin.ac.jp (T.M.); narusek@dpc.agu.ac.jp (K.N.); 3Department of Clinical Laboratory, Aichi Gakuin University Dental Hospital, Nagoya 464-8651, Japan; minato@dpc.agu.ac.jp; 4Diabetic Neuropathy Project, Department of Diseases and Infection, Tokyo Metropolitan Institute of Medical Science, Tokyo 156-8506, Japan; sango-kz@igakuken.or.jp

**Keywords:** Schwann cells, insulin, myelin protein zero, myelin basic protein, myelin associated glycoprotein, Akt, extracellular-signal-regulated kinase (ERK)

## Abstract

Schwann cells play an important role in peripheral nerve function, and their dysfunction has been implicated in the pathogenesis of diabetic neuropathy and other demyelinating diseases. The physiological functions of insulin in Schwann cells remain unclear and therefore define the aim of this study. By using immortalized adult Fischer rat Schwann cells (IFRS1), we investigated the mechanism of the stimulating effects of insulin on the cell proliferation and expression of myelin proteins (myelin protein zero (MPZ) and myelin basic protein (MBP). The application of insulin to IFRS1 cells increased the proliferative activity and induced phosphorylation of Akt and ERK, but not P38-MAPK. The proliferative potential of insulin-stimulated IFRS1 was significantly suppressed by the addition of LY294002, a PI3 kinase inhibitor. The insulin-stimulated increase in MPZ expression was significantly suppressed by the addition of PD98059, a MEK inhibitor. Furthermore, insulin-increased MBP expression was significantly suppressed by the addition of LY294002. These findings suggest that both PI3-K/Akt and ERK/MEK pathways are involved in insulin-induced cell growth and upregulation of MPZ and MBP in IFRS1 Schwann cells.

## 1. Introduction

Diabetic peripheral neuropathy (DPN) is one of the complications of diabetes mellitus. It is reported to be caused by many factors, including abnormalities in polyol metabolic pathways, oxidative stress, and ion channel abnormalities, but the details of these causes remain unclear [1,2,3].

Although the etiology of DPN is clearly related to hyperglycemia, there are reports that insulin resistance exacerbates the pathogenesis of diabetic neuropathy in type 2 diabetes mellitus, and impaired insulin signaling in the peripheral nerves is thought to play a major role in the pathogenesis of diabetic neuropathy [4]. These results suggest the pivotal role of insulin in peripheral nerve system (PNS) function. However, many details remain unclear.

Several animal studies showed that a loss of insulin signaling in PNS may contribute to DPN [5,6]. Similarly, some reports indicated that low-dose insulin, which is insufficient in reducing hyperglycemia, may have a beneficial effect on the signs and symptoms of DPN [7,8,9]. Insulin treatment shows the beneficial effects on DPN-related sensory deficits, as well as late-onset diabetes-induced motor neuropathy [10].

Schwann cells are the principal glia of PNS, which function to support neurons in the PNS. Schwann cells are derived from neural crest cells, and there are two types of Schwann cells—myelinating and non-myelinating [11,12]. Myelinating Schwann cells wrap around the axons of motor and sensory neurons to form the myelin sheath. Proteins such as myelin protein zero (MP0), myelin basic protein (MBP), and myelin-associated glycoprotein (MAG) are known to be involved in the formation of the myelin sheath [12,13]. MP0 is a transmembrane protein constituting the majority of the protein in the myelin sheath [14]. Schwann cell dysfunction is related to diabetic neuropathy [15,16,17].

Several studies have reported that the abnormal production of Schwann constituent proteins is promoted by reduced insulin signaling [18,19]. However, sufficient investigations into the direct effects of insulin on Schwann cells, independent of glycemic control, have not yet been undertaken. Despite the progress of in vivo studies of insulin signaling in nerve fibers [7], there are few reports that examine the direct action of insulin on cultured Schwann cells with regard to how insulin promotes myelin-constituent proteins that do not depend on glycemic control [18].

Immortalized rat Schwann cells (IFRS1) are the Schwann cell line obtained from the peripheral nerve tissue of Fischer 344 rats. IFRS1 shows the characteristics of mature Schwann cells, such as the expression of neurotrophic factors and myelin proteins and the myelination in co-culture with neuronal cells [20,21,22,23]. IFRS1 is a valuable tool for studying neuron–Schwann cell interactions, as well as the pathology of the axonal degeneration and regeneration in PNS. In this study, we investigated the mechanism by which insulin promotes the growth and expression of myelin component proteins in cultured Schwann cells using IFRS1.

## 2. Results

### 2.1. Identification of Insulin Receptors

To examine whether IFRS1 has insulin receptors, we evaluated the expression of insulin receptor beta (IR-beta). Examination of IFRS1 under an optical microscope revealed that the cells were homogeneous (Figure 1A). IR-beta was observed by immunofluorescence staining in IFRS1 (Figure 1B). Using Western blot analysis, IR-beta was identified as a single band at 95 kDa (Figure 1C).

### 2.2. Insulin Stimulates the Phosphorylation of Akt, MEK, and Extracellular-Signal-Regulated Kinase (ERK) in IFRS1

In Western blot analyses, insulin stimulated the phosphorylation of Akt, MEK, and ERK over time, and a phosphorylation peaks was observed at 5, 10, and 10 min after each stimulation (Figure 2A–C). In addition, phosphorylation of Akt, MEK, and ERK was observed in a dose-dependent manner, and a peak was observed at a concentration of 1 μM (Figure 2E–G). On the other hand, insulin did not stimulate the phosphorylation of p-38 MAPK in IFRS1 (Figure 2D,H).

### 2.3. Insulin Promotes the Proliferation of IFRS1

Cell proliferation was assessed using an MTT assay and CCK8. Insulin significantly increased the proliferation of IFRS1, as shown by the twofold increase from that of the control in the MTT assay (2.0 ± 0.2-fold from the control; Figure 3A). These effects were inhibited by LY294002 at 3 μM (2.0 ± 0.2-fold vs. 1.6 ± 0.2-fold from the control; *p* = 0.008). Incidentally, more than 3μM of LY294002 inhibited the phosphorylation of Akt and more than 30 nM of PD98059 inhibited the phosphorylation of ERK (data not shown). Insulin-stimulated cell proliferation was not inhibited by PD98059 in the MTT assay. Similarly, insulin increased the proliferation of IFRS1 through a 2.2-fold increase in CCK8 (2.2 ± 0.3-fold from the control; Figure 3B). These effects were inhibited by LY294002 at 3 μM (2.2 ± 0.3-fold vs. 1.6 ± 0.2-fold from the control; *p* = 0.007)), suggesting that insulin stimulates cell proliferation via PI3-K/Akt. Similarly, insulin-stimulated cell proliferation was not inhibited by PD98059 in CCK8.

### 2.4. Insulin Promotes the MPZ Expressions in IFRS1 via MEK/ERK

The long-term stimulation of insulin increased the protein expression of myelin protein MPZ. PD 98059 was shown to inhibit the increase in the expression of MPZ by insulin, but not LY294002 (Figure 4A). These trends were also confirmed by fluorescence microscopy (Figure 4B).

### 2.5. Insulin Promotes the MBP Expressions in IFRS1 via PI3-K/Akt

Insulin significantly increased the expression of MBP. LY294002 was shown to inhibit an increase in the expression of MBP through insulin (Figure 5A). This trend was also confirmed by fluorescence microscopy (Figure 5B). PD98059 did not affect the insulin-stimulated MBP expression. On the other hand, insulin did not increase the expression of MAG (Appendix A). The inhibition of Akt and ERK showed no additional effects.

## 3. Discussion

DPN is the most frequent complication in patients with diabetes. Many causes of the etiology of PDN are suggested, but the details are still unclear. Several studies have reported that insulin dysregulation contributes to neuropathic changes in sensory neurons [24,25]. Most of the studies on insulin signaling in peripheral nerves have focused primarily on DRG neurons, and reports on the direct effects of insulin on Schwann cells alone and the resulting insulin signaling are rare [18]. In this study, we demonstrated, for the first time, the mechanism by which direct stimulation of insulin promotes the proliferation and expression of myelin component proteins using IFRS1, and revealed the involvement of the PI3-K/Akt pathway in the insulin-stimulated proliferation and MBP production, as well as the MEK/ERK pathway in the insulin-stimulated MPZ production in IFRS1.

The myelin sheath is a multilayered membrane unique to the nervous system and serves as an insulator that greatly increases the speed of the axonal impulse conduction [26]. MPZ, which is absent in the central nervous system, is a monolayer glycoprotein, and is most abundant in the myelin sheath of peripheral nerves, accounting for more than half of the total. MPZ has been reported to play an important role in the formation and maintenance of myelin [26]. MPZ mutations are closely associated with diseases such as Charcot–Marie–Tooth neuropathy type 1B, Dejerine–Sottas syndrome, and congenital hypo-myelinating neuropathy [27,28,29]. In diabetic conditions, Cermenati et al. reported that MPZ decreases for diabetogenic rats with streptozotocin [30].

In oligodendrocytes, Jeffries et al. reported that myelin thickness is increased by Akt-independent ERK 1/2 activation [31]. In this study, PD98059 treatment in Schwann cells, a major component of PNS, suppressed the increased protein expression of MPZ [32]. We also showed that insulin promoted the MPZ expression via MEK/ERK in cultured Schwann cells, but not PI3-K/Akt. The MEK/ERK signaling cascade is activated by various types of receptor tyrosine kinases involved in cell proliferation and differentiation, including insulin receptors [33]. Many studies have reported that the MEK/ERK signaling cascade is important for the maintenance of PNS function, and our study suggests that this may be the case. On the other hand, p38-MAPK activation promotes the denervated Schwann cell phenotype and functions as a negative regulator of Schwann cell differentiation and myelination [34]. The involvement of p38-MAPK through direct insulin stimulation was not observed in IFRS1, which is in agreement with previous reports.

MBP is a key protein in the process of myelination in the nervous system, acting as an insulator that greatly increases the rate of axonal impulse conduction. MBP interacts with lipids in the myelin membrane to maintain the correct structure of myelin, interacting with lipids in the myelin structure [35]. Several cytokines are involved in the PI3-K/Akt-mediated promotion of the MBP expression, but there are no reports that direct insulin stimulation promotes MBP expression via PI3K/Akt in cultured Schwann cells. This study has provided a new finding with regard to Schwann cells. Several reports have shown that MPZ is associated with Akt and MBP is associated with ERK [36]. These associations may be a unique result of the direct insulin stimulation of IFRS1.

Myelin-associated glycoprotein (MAG; Siglec-4) is a transmembrane protein glycoprotein that localizes to periaxonal Schwann cell and oligodendrocyte membranes, and is involved in glial–axonal interactions. MAG is thought to be involved in myelination during neuronal regeneration in the nervous system. MAG is thought to be involved in myelination during nerve regeneration in the nervous system, and is essential for the long-term survival of myelinated axons following myelinogenesis [37,38]. Rachana et al. reported that the expression of MBP and MAG in Schwann cells from the sciatic nerve of puppies tended to be reduced in hyperglycemic conditions and improved with insulin treatment [18,39]. In this study, direct insulin stimulation did not promote the protein expression of MAG, nor did various inhibitors suppress its expression, which are not in agreement with ex vivo studies of MAG. This result reflects the limitations of myelin formation through only insulin under physiological conditions, and there are limits to the therapeutic potential of insulin alone for DPN.

We showed that two intracellular signals, PI3-K/Akt and MEK/ERK, are important for the myelin functions. These results indicate that insulin may be beneficial for Schwann cell repair, and are consistent with previous reports that insulin stimulation promotes myelin-forming protein synthesis [5,18,39].

The results with regards to the proliferative capacity showed that insulin stimulation not only promoted the expression of the myelin protein, but also increased cell proliferation itself. In addition, LY294002, an inhibitor of PI3-K, suppressed the increase in proliferation induced by insulin stimulation in both MTT and CCK8 assays, indicating that insulin promotes proliferation via PI3-K/Akt.

IFRS1 is relatively monoclonal and stable. IFRS1 seems to be the appropriate cell to study the proliferative potential, indicating that monoclonal cultured Schwann cells purely reflect insulin signaling. It is interesting to note that this is the first time that we have shown that insulin promotes the proliferative potential of IFRS1 through PI3-K/Akt and not through MEK/ERK. PI3-K, located upstream of Akt, is one of the most important regulatory proteins, controlling important functions such as cell growth, senescence, and transformation [40]. Akt is an important regulator of various cellular processes, including glucose metabolism and cell survival [41,42,43].

Other reports show that the activation of PI3-K through axonal factors promotes Schwann cell proliferation and survival, and implicate PI3-K in the early events of myelination [44].

This study supports the results of other studies suggesting that the activation of PI3-K/Akt promotes Schwann cell proliferation and survival, and that PI3-K is involved in the major events of myelination.

There are several limitations to this study. We were not able to conclude whether endogenous insulin is required as a driver of myelination or whether it is physiologically important. We were not able to analyze the role of Akt and ERK in insulin receptor-mediated myelination in vivo. We did not compare of the proliferative potential between primary Schwann cells and IFRS1. Furthermore, we failed to investigate the axon-glia interaction that should underlie myelination and myelin repair. Further studies on this issue are needed, such as co-culture with PC12 cells and neurons such as DRGs.

In this experiment, we focused on the upstream of insulin signaling to investigate the mechanism of promoting viability and myelin protein expression, and showed that both PI3-K/Akt and MEK/ERK pathways are important for myelin formation (Figure 6). These results suggest the crucial role of insulin signaling in the pathogenesis of Schwann cells, the key player in peripheral nerves.

## 4. Materials and Methods

### 4.1. Cell Culture

IFRS1 were grown in Dulbecco’s Modified Eagle Medium (DMEM) containing 10% FBS, 20 ng/mL neuregulin-beta (heregulin-beta; Upstate, Darmstadt, Germany), 2 μM forskolin (Sigma-Aldrich, Darmstadt, Germany), and dimethyl sulfoxide (DMSO) at 37 °C in a humidified 5% CO2/95% air atmosphere [15,23]. To investigate the various effects of insulin stimulation, the cells were serum-starved overnight and insulin was stimulated for 5, 10, 30, and 60 min at a concentration of 100 nM.

In order to investigate the long-term effects of stimulation, it was replaced with DMEM containing 50 μg/mL ascorbic acid (Lonza, Basel, Switzerland) and stimulated for 5 days at a concentration of 100 nM insulin. Stimulation was carried out on days 0, 2, and 4, and the protein was recovered on day 5.

LY 294002 (Sigma-Aldrich, Darmstadt, Germany) and PD 98059 (Sigma-Aldrich, Darmstadt, Germany) were applied 30 min before stimulation with insulin.

### 4.2. Immunofluorescence Staining of IFRS1

Cultured cells were fixed with 4% paraformaldehyde in 0.1 M PBS; permeabilized with 0.05%-Triton X-100 in PBS; and stained with anti-phospho Akt antibody, anti-phospho ERK antibody (Cell Signaling Technology; Beverly, MA, USA), and Alexa Fluor 568-conjugated phalloidin (Invitrogen CO, USA; Carlsbad, CA, USA). Fluorescence was examined using a confocal fluorescence microscope (FV100, Olympus; Tokyo, Japan).

### 4.3. Proliferation Analyses of IFRS1

Cells were seeded in a 96-well plate in 100 μL per well of a cell suspension of 1.0 × 10^5^ cells/mL. The cell growth was arrested using serum-free DMEM for 12 h. After pre-incubation with the phosphatidylinositol 3-kinase (PI3-K) inhibitor LY294002 (1 μM) or the mitogen-activated protein kinase/ERK kinase (MEK) inhibitor PD98059 (5 μM), the cells were incubated with insulin (100 nM). MTT (3-(4,5-dimethylthiazol-2-yl) -2,5-diphenyltetrazolium bromide; Sigma-Aldrich) was added at a final concentration of 0.5 mg/mL, and after a further 2 h of incubation, IFRS1 was lysed with isopropanol containing 0.04 M HCl. The MTT reduction was read at 550 nm using a spectrophotometer (ARVO MX, Perkin Elmer Cetus; Norwalk, CT). The proliferation activity was measured 24 h later. For measurement with CCK8, 10 μL per well of Cell Counting Kit-8 (Dojindo Laboratories, 347-07621, Kumamoto, Japan) were added and reacted at 37 °C under a humidified condition for 4 h, and the absorbance was measured at 450 nm. In order to investigate the influence of the PI3-K/Akt and MEK/ERK pathway on insulin-stimulated growth, DMEM containing LY294002 (concentration 300 nM/3 μM/30 μM), SB 203580 (concentration 300 nM/3 μM/30 μM), or PD 98059 (concentration 300 nM/3 μM/30 μM) was administered for 30 min after the stimulation with insulin was carried out.

### 4.4. Western Blot Analyses

Cell lysates were prepared with a cell lysis buffer (Cell Signaling #9803, Beverly, MA, USA). After incubation with each experimental medium for the indicated periods, the cells were washed three times with ice-cold PBS and lysed in a Cell Lysis Buffer (Cell Signaling #9803, Beverly, MA, USA) with 1% phenylmethylsulphonyl fluoride (Sigma-Aldrich 93482, Darmstadt, Germany). A cell solution containing the same amount of protein was mixed with 4× Laemmli Sample Buffer (Bio-rad 1610747, Hercules, CA, USA) at a ratio of 3:1 and heated at 100 °C for 5 min to prepare the sample. Samples containing the same amount of protein were electrophoresed on SDS-PAGE and transferred to a nitrocellulose membrane. The membrane was incubated overnight at 4 °C with the first antibody, followed by incubation with an HRP-conjugated anti-rabbit polyclonal IgG antibody. The proteins were visualized using ECL chemiluminescence detection kits [45].

### 4.5. Identification of Insulin Receptor in IFRS1

To identify the insulin receptor in IFRS1, confluent cultures of IFRS1 were lysed with a lysis buffer, subjected to SDS-PAGE, and detected using anti-human IR-beta antibody (4B8) Rabbit mAb (Cell Signaling #3025, Beverly, MA, USA).

### 4.6. Statistical Analysis

All of the group values were expressed as the means ± SEM. Statistical analyses were performed using one-way ANOVA with the Bonferroni correction for multiple comparisons. The level of significance was set at *p* < 0.05 (SPSS ver.22.0, IBM; Armonk, NY, USA).

## Figures and Tables

**Figure 1 ijms-22-05505-f001:**
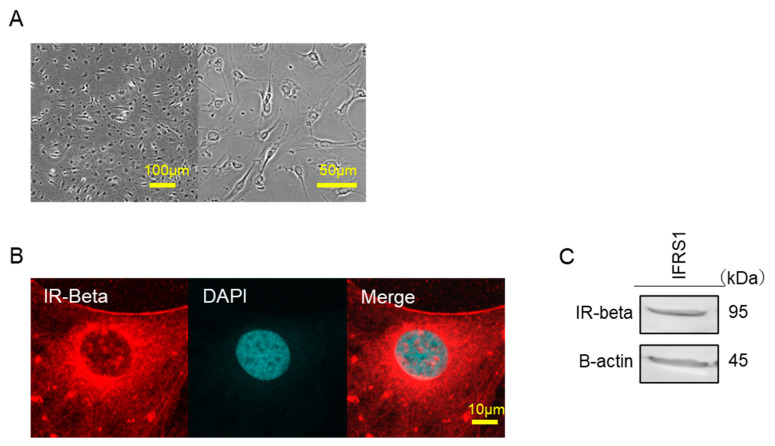
(**A**) Differential interference contrast (DIC) microscopic image of IFRS1. (**B**) Confocal immunofluorescence analysis of IFRS1 treated with anti-IR-beta rabbit mAb labeled with Alexa Fluor 568-conjugated phalloidin (red). Blue pseudo color-DAPI. (**C**) IR-beta was identified by Western blot analyses using the anti-IR-beta antibody.

**Figure 2 ijms-22-05505-f002:**
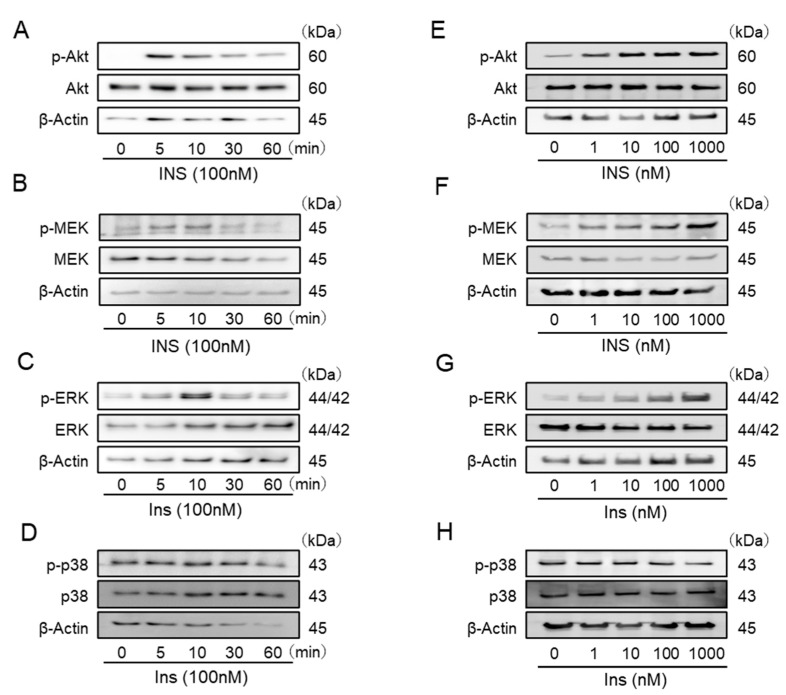
(**A**–**D**): The phosphorylation of Akt, MEK, extracellular signal-regulated kinase (ERK), and 38-mitogen-activated protein kinase (MAPK) by insulin. Insulin stimulation (100 nM) was conducted for the indicated time. (**E**–**H**): The concentration-dependent phosphorylation of Akt, MEK, ERK, and p38-MAPK by insulin.

**Figure 3 ijms-22-05505-f003:**
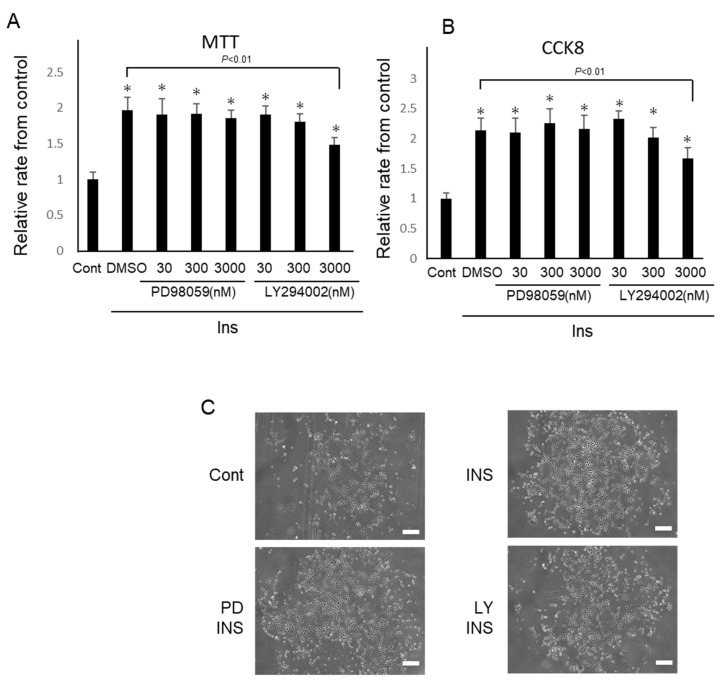
Insulin promotes the proliferation of IFRS1. Cell growth was arrested using serum-free Dulbecco’s Modified Eagle Medium (DMEM) for 12 h. After pre-incubation with LY294002 (30–3000 nM) or PD98059 (30–3000 nM), cells were incubated with insulin (100 nM). (**A**) Cell proliferation was assessed using an MTT assay. (**B**) Cell proliferation was assessed by CCK-8. The results are shown as the mean ± standard error of the mean (SEM). * *p* < 0.01 versus the control. (**C**) Representative images of the IFRS1 observed under phase-contrast microscopy. (Bar = 200 μm).

**Figure 4 ijms-22-05505-f004:**
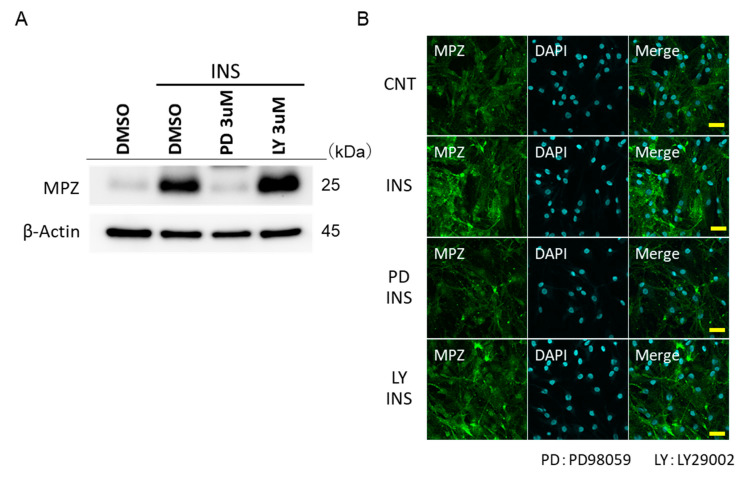
Data on insulin-stimulated myelin protein zero (MPZ) expression through various inhibitors. (**A**) Western blot data showing the expression of MPZ through insulin stimulation, and the effect of LY294002 and PD98059 on MPZ. The results of one of three experiments with similar results are shown. * *p* < 0.05 versus insulin (DMSO control). (**B**) Immunofluorescence-stained image of MPZ. Green indicates MPZ and blue indicates DAPI (Bar = 100 μm).

**Figure 5 ijms-22-05505-f005:**
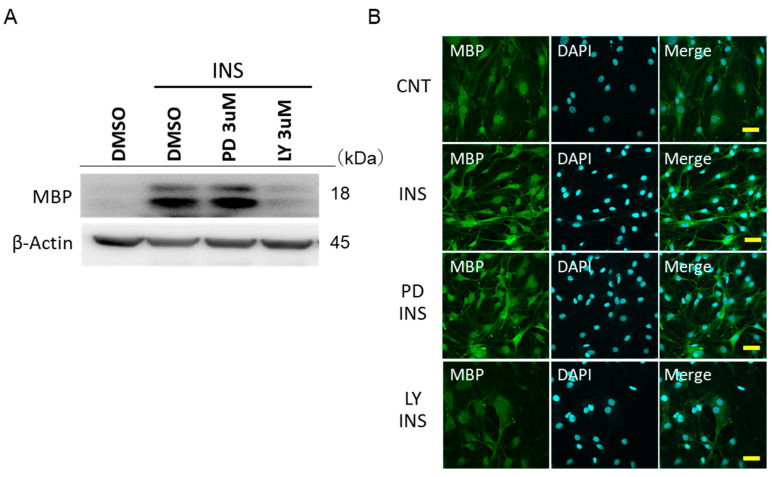
Data on insulin-stimulated myelin basic protein (MBP) expression by various inhibitors. (**A**) Western blot data showing the expression of MBP by insulin stimulation and the effect of LY294002 and PD98059 on MBP. The results of one of three experiments with similar results are shown. * *p* < 0.05 versus insulin (DMSO control). (**B**) Immunofluorescence-stained image of MBP. Green indicates MBP and blue indicates DAPI (Bar = 100 μm).

**Figure 6 ijms-22-05505-f006:**
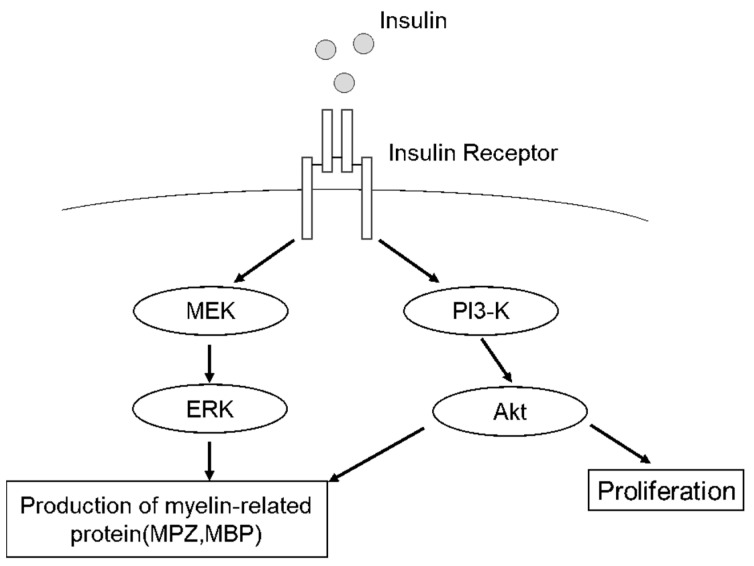
Proposed scheme for insulin-stimulated signaling of the proliferation and the production of myelin-related proteins. Insulin promotes the MPZ expression of IFRS1 via MEK/ERK and MBP expression of PI3-K/Akt, whereas insulin promotes the proliferation of IFRS1 via PI3-K/Akt.

## Data Availability

N/A.

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
