# Peer review of "The Effects of Insulin on Immortalized Rat Schwann Cells, IFRS1"

_ijms, 2021, doi:10.3390/ijms22115505_

Round 1
Reviewer 1 Report
In the manuscript by Saiki et al., the authors seek to address the role of insulin on peripheral nervous system glia - Schwann cells - to elucidate mechanisms of diabetic peripheral neuropathy (DPN).
Firstly, I want to point out that the title of the study is incorrect as “myelin formation”, as per its definition, is by no means limited to the expression of myelin protein zero but in fact requires a substrate on which said myelin is formed and deposited (in the case of Schwann cells, it has to be neuronal projections). I could not find, in any of the 5 figures included in the manuscript, an analysis of myelin formation in response to insulin.
Secondly, it is not clear anywhere in the manuscript how an analysis of cell proliferation and signaling pathways that positively regulate myelin biogenesis is relevant to speculate about the peripheral glia contribution to DPN, which affects adults. While both (proliferation, myelination potential) are aspects crucial for developmental myelination, its underlying mechanistics do not directly translate to the adult repair Schwann cell paradigm.
Thirdly, the foundation of myelination and myelin repair is the axon:glia interaction unit. The authors have to use extra-caution when postulating physiologically relevant conclusions – be it about the developmental Schwann cell or the adult repair Schwann cell – from an immortalized cell line cultured alone.
To significantly improve the value of the work for the field, I suggest that the authors reevaluate the conclusions that can be confidently drawn from the data present and reorient the main message of the manuscript.
Author Response
We are grateful to reviewers for the critical comments and useful suggestions that have helped us improve our paper. As indicated in our responses, we have taken all of the comments and suggestions into account in the revised version of our paper.
1. Firstly, I want to point out that the title of the study is incorrect as “myelin formation”, as per its definition, is by no means limited to the expression of myelin protein zero but in fact requires a substrate on which said myelin is formed and deposited (in the case of Schwann cells, it has to be neuronal projections). I could not find, in any of the 5 figures included in the manuscript, an analysis of myelin formation in response to insulin.
We apologize that the title of the study is incorrect as "myelin formation". We have replaced the title to be better for represent this study (line 2-3 in page 1). We have corrected all terms of myelination about our experiment in this manuscript (line 141-142 in page7).
2 Secondly, it is not clear anywhere in the manuscript how an analysis of cell proliferation and signaling pathways that positively regulate myelin biogenesis is relevant to speculate about the peripheral glia contribution to DPN, which affects adults. While both (proliferation, myelination potential) are aspects crucial for developmental myelination, its underlying mechanistics do not directly translate to the adult repair Schwann cell paradigm.
As the reviewer pointed out, it showed how insulin is physiologically involved in myelin formation and did not reflect the repair mechanism of Schwann cells in adults. We have described the information as study limitation in this revised manuscript (line 229-230 in page 9).
3. Thirdly, the foundation of myelination and myelin repair is the axon:glia interaction unit. The authors have to use extra-caution when postulating physiologically relevant conclusions – be it about the developmental Schwann cell or the adult repair Schwann cell – from an immortalized cell line cultured alone.
We appreciate the reviewer’s comment. As well as the reviewer’s comment 2, we have incorporated the sentence “We failed to investigate the axon-glia interaction that should underlie myelination and myelin repair.” in the revised manuscript (line 230-231, page 9).
Reviewer 2 Report
This is a very comprehensive study concerning the insulin to increase the proliferative activity in IFRS1 cells by inducing the phosphorylation of Akt and ERK pathway. Even though the IFRS1 cells are relatively monoclonal and stable, these immortalized cells are different from the Schwann cells. For the increased power of science, in the Figure 3, the authors should add the data of primary Schwann cells culture in the MTT and CCK 8 assays.
Author Response
We are grateful to reviewers for the critical comments and useful suggestions that have helped us improve our paper. As indicated in our responses, we have taken all of the comments and suggestions into account in the revised version of our paper.
This is a very comprehensive study concerning the insulin to increase the proliferative activity in IFRS1 cells by inducing the phosphorylation of Akt and ERK pathway. Even though the IFRS1 cells are relatively monoclonal and stable, these immortalized cells are different from the Schwann cells. For the increased power of science, in the Figure 3, the authors should add the data of primary Schwann cells culture in the MTT and CCK 8 assays.
We are grateful to reviewers for the useful suggestions that have helped us improve our paper. Although, IFRS1 shows the characteristics of mature Schwann cells, such as the expression of neurotrophic factors and myelin proteins and the myelination in co-culture with neuronal cells, we agreed that one or more experiments of the primary Schwann cells would greatly strengthen the impact of this manuscript. We apologize that we failed to add the additional experimental data of primary Schwann cells culture in the MTT and CCK 8 assays in a short period of time. We have incorporated the sentence “We did not compare of the proliferative potential between primary Schwann cells and IFRS1.” in the revised manuscript (line 229-230, page 9). We intend to further examine the differences between primary Schwann cells and IFRS1in co-culture with the neuron (e.g.: DRG).
Reviewer 3 Report
In this study, the authors aim to determine the physiological function of insulin on Schwann cells. The authors reported that application of insulin to IFRS1 cells increased the proliferative activity and induced phosphorylation of Akt and ERK, but not P38-MAPK. Also, the proliferative potential of insulin-stimulated IFRS1 was significantly suppressed by the addition of LY294002, a PI3 kinase inhibitor. Additionally, the insulin-stimulated increase in MPZ expression was significantly suppressed by the addition of PD98059, a MEK inhibitor. Furthermore, insulin-increased MBP expression was significantly suppressed by the addition of LY294002. Finally, the authors concluded that both PI3-K/Akt and ERK/MEK pathways are involved in insulin-induced cell growth and up-regulation of MPZ and MBP in IFRS1 Schwann cells. The authors’ results generally support the conclusion and I only have one minor comment that the language use in this manuscript requires some revision. For example, even in the abstract, “On the other hand, and insulin-increased MBP expression was significantly suppressed by the addition of LY294002” should be “Furthermore, insulin-increased MBP expression was significantly suppressed by the addition of LY294002.” This manuscript is highly recommended to be read and edited by a native English speaker before publication.
Author Response
We are grateful to reviewers for the critical comments and useful suggestions that have helped us improve our paper. As indicated in our responses, we have taken all of the comments and suggestions into account in the revised version of our paper.
The authors’ results generally support the conclusion and I only have one minor comment that the language use in this manuscript requires some revision. For example, even in the abstract, “On the other hand, and insulin-increased MBP expression was significantly suppressed by the addition of LY294002” should be “Furthermore, insulin-increased MBP expression was significantly suppressed by the addition of LY294002.” This manuscript is highly recommended to be read and edited by a native English speaker before publication.
We appreciate the reviewer’s comment. As the reviewer indicated, an English native speaker of the MDPI English editing service has re-corrected the revised manuscript. We have changed the conjunction in the abstract (line 26 in page 1) and have changed the sentence “sufficient investigations into the direct effects of insulin on Schwann cells, independent of glycemic control, have not yet been undertaken.” (line 59-61, page 2). Minor points were changed. (line 36, 37, 44, 46, 56, 64, 78, 79, 89, 90, 99, 100, 152, 178, 183, 186, 188, 200, 207, 215)